# Sexual and reproductive health service utilization among adolescent girls in Kenya: A cross-sectional analysis

Lonnie Embleton[1], Paula Braitstein[2,3,4], Erica Di Ruggiero[1]*, Clement Oduor[5], Yohannes Dibaba Wado[5]

1 Dalla Lana School of Public Health, Centre for Global Health, University of Toronto, Toronto, Ontario, Canada, 2 Department of Epidemiology, Dalla Lana School of Public Health, University of Toronto, Toronto, Ontario, Canada, 3 College of Health Sciences, School of Medicine, Moi University, Eldoret, Kenya, 4 Academic Model Providing Access to Healthcare, Eldoret, Kenya, 5 African Population and Health Research Center, Nairobi, Kenya

* e.diruggiero@utoronto.ca

**Data Availability Statement:** The data is available in a public repository at APHRC. The data from this analysis can be accessed by submitting a data

## Abstract

We examined the association between adolescents' sexual and reproductive health (SRH) service utilization in the past 12 months and structural, health facility, community, interpersonal, and individual level factors in Kenya. This cross-sectional analysis used baseline data collected in Homa Bay and Narok counties as part of the In Their Hands intervention evaluation from September to October 2018. In total, 1840 adolescent girls aged 15 to 19 years were recruited to complete a baseline survey. We used unadjusted and adjusted logistic regression to model factors associated with SRH utilization across the social-ecological framework levels. Overall, 36% of participants reported visiting a health facility for SRH services in the past 12 months. At the structural level being out-of-school (AOR: 2.12 95% CI: 1.60–2.82) and not needing to get permission to go (AOR: 1.37 95%CI: 1.04–1.82) were associated with SRH service utilization. At the interpersonal level, participants who reported being able to ask adults for help when they needed it were more likely to report using SRH services in the past 12 months (AOR: 1.98, 95% CI: 1.09–3.78). At the individual level, having knowledge about where to obtain family planning (AOR = 2.48 95% CI: 1.74–3.57) and receiving information on SRH services in the past year (AOR: 1.44 95% CI:1.15–1.80) were associated with SRH service utilization. Our findings demonstrate the need for interventions, policies, and practices to be implemented across structural, health facility, community, interpersonal, and individual levels to comprehensively support adolescent girls to access and use SRH services.

## Introduction

Adolescents aged 10 to 19 years in Kenya are an underserved population in the health system, yet represent a large proportion of the population (24%) [1,2]. Adolescent girls, in particular, need access to high-quality adolescent and youth-friendly health services (AYFS), as they experience a substantial burden of adverse sexual and reproductive health (SRH) outcomes [2–4].

request in the APHRC Micro-data portal https://aphrc.org/microdata-portal.

**Funding:** The evaluation of the In Their Hands (ITH) Project is funded by Children's Investment Fund Foundation (CIFF) with a grant to the African Population and Health Research Center (Grant reference number: R-1710-02085). The funders had no role in study design, data collection and analysis, decision to publish, or preparation of the manuscript. The authors received no specific funding for this publication.

**Competing interests:** The authors have declared that no competing interests exist.

Adolescent girls in Kenya have high rates of unintended pregnancy [5–7], undergo unsafe abortions [2,8], experience sexual abuse and violence [9], and disproportionately acquire sexually transmitted infections (STIs) and human immunodeficiency virus (HIV) [3,10–12], contributing to poor sexual health.

Sexual health, defined as the attainment of physical, emotional, mental, and social well-being related to sexuality [13], is crucial to achieving health for all, in line with the Sustainable Development Goals (SDG 3) [14]. The absence of good sexual health for adolescent girls can have long-term consequences, not only for their overall health and well-being, but for their social and economic circumstances into adulthood. Ensuring universal health coverage (SDG 3.8) and access to SRH services (SDG 3.7) [14], are fundamental components of attaining sexual health for adolescents and securing their futures. To avert poor SRH outcomes, adolescent girls require access to comprehensive SRH services [15].

Across sub-Saharan African countries, adolescents experience structural, health facility, community, interpersonal, and individual level barriers and facilitators to accessing and utilizing SRH services [2,16–18]. For instance, at the structural level, adolescent girls may encounter barriers to care such as laws and policies that require parental or partner consent [2,16]. Health facilities may lack dedicated adolescent-friendly spaces or have staff and providers that stigmatize and discriminate against adolescents [16,17,19,20]. At the community and interpersonal levels, social-cultural norms often construct adolescent sexuality as taboo, preventing adolescent girls from utilizing SRH services for fear of judgement and discrimination from parents, peers, and other community members [21–24]. Finally, individual level factors such as self-efficacy, agency, SRH knowledge, and awareness about the availability and type of services offered may all act as barriers or facilitators to SRH utilization among adolescent girls [2,16].

Kenya has a robust National Adolescent SRH Policy and National Guidelines for the Provision of AYFS applicable to all tiers of the health system [2,25]. The AYFS guidelines include an essential package of SRH services and four service delivery models, including community-, clinical-, school-, and virtual-based services. The range of service delivery models aims to support engaging Kenya's diverse population of adolescents in SRH services who live in different geographic regions of the country with varied health and socioeconomic statuses [2]. Despite the well-developed national policies and guidelines for the provision of AYFS, it remains unclear if adolescent girls in Kenya are accessing and utilizing SRH services. To fill this gap in knowledge, this analysis investigated SRH service utilization among adolescent girls aged 15–19 years residing in Homa Bay and Narok counties. More specifically, we examined the association between adolescent SRH service utilization in the past 12 months and structural, health facility, community, interpersonal, and individual level factors. We sought to identify key barriers and facilitators at each of these levels that may be targeted to improve access to and utilization of SRH services for adolescents in this context.

## Methods

### Study design

This cross-sectional analysis used baseline data from the In Their Hands (ITH) intervention evaluation [5,21], which used a before and after mixed methods design. The ITH intervention sought to increase adolescents' use of SRH services through a digital platform that linked girls to SRH services across 18 counties. Two counties, Homa Bay and Narok, were purposively selected for the evaluation, as the ITH intervention had not yet begun in these counties at the start of the evaluation. At baseline, a cross-sectional survey was administered between September 1st, 2018, to October 12th, 2018, to adolescent girls aged 15–19 years in Homa Bay and Narok counties. The present analysis uses this cross-sectional survey data.

## Study setting

Homa Bay is located in southwestern Kenya on Lake Victoria with a population of 1,131,950. Adolescents aged 10–19 years make up 28% (318,121) of the population. In 2014, Homa Bay had one of the highest adolescent pregnancy rates in the country (33%), following Narok county (40% for adolescents aged 15–19 years [26]. Narok county is located in the southern part of the Rift Valley with a population of 1,157,873, and adolescents aged 10–19 years make up 26% (304,240) of the population [27].

## Study participants

Adolescent girls were eligible to participate in the survey if they were between the ages of 15–19 years, were a resident in the study area for at least six months and were living in the sampled households. Adolescent girls who were students and residing in boarding schools were excluded from the study.

## Sample size, sampling, and recruitment

Using Cochran's formula, a sample size of 1885 was calculated to detect differences in participants' intention to use contraception in the future between baseline and endline. The study team conducted community mobilization and sensitization in the study sites before any household listing and data collection activities. Supervisors and team leaders led community engagement at the village level and obtained approvals from community leadership to carry out the survey in the site. Village elders and local guides supported the team when entering communities and locating households. At the county and sub-county levels, the research team organized meetings with county commissioners and county health. Sampling lists were then created based on a process that identified all eligible adolescent girls residing in households in select villages within health facility catchment areas within a 10km radius from the health facility) affiliated with the intervention. Within each county, 3 sub-counties were sampled, which included West Kasipul, Ndhiwa and Kasipul in Homa Bay, and Narok Town, Narok South and Narok East in Narok county. In each sub-county, three wards were selected, and villages were selected within the wards. In total, 22 villages in Narok and 24 villages in Homa Bay were sampled. In each of the villages, household sampling lists were generated where adolescent girls resided. A total of 1897 households with eligible respondents were identified. One adolescent girl was randomly selected per household to participate in the survey by the SurveyCTO program. The response rate for participation was 97%. In total, 1840 adolescent girls were successfully recruited and participated in the cross-sectional survey; 57 adolescent girls selected did not participate due to lack of parental consent, unavailability, or refusal to participate.

## Ethical considerations

This study received approval from the Amref Health Africa Ethics and Scientific Review Committee (AMREF-ESRC P499/2018) and a research permit from the National Commission for Science, Technology, and Innovation (NACOSTI). Additional approvals were sought from local administrators and Ministries of Health and Education in the respective counties where the study was conducted. Parents/guardians provided written informed consent for participants younger than 18 years and adolescents under 18 provided written informed assent to participate. Married and other emancipated minors and those aged 18 years or older provided written informed consent to participate.

## Study procedures

A team of 17 research assistants were trained for five days on the study protocol, survey tool, and the informed consent process before conducting data collection. The research assistants worked with local leaders (Area Chiefs, Assistant Chiefs, and Village Elders), and community health volunteers to mobilize study participants. Research assistants visited all households and provided information about the study to the household heads. Consent was sought from the household heads to be part of the study and to be included in the household listing. One eligible adolescent was randomly selected per household. After receiving informed consent and assent for adolescent girls' participation, the research assistants conducted the survey face-to-face using tablets programmed with the questionnaire.

## Data and measures

The survey was developed in English and translated into Kiswahili. The questionnaire was programmed into the ODK-based Survey CTO platform for data collection and management. The survey was pre-tested with a sample of 42 adolescent girls aged 15–19 years in the Korogocho informal settlement in Nairobi. The survey collected data on socio-demographics, access to and use of media, sources of information for SRH, family planning knowledge and access, social networks, self-efficacy, self-esteem, and agency, sexual activity, contraceptive use, and use of SRH services and quality of care. The primary outcome of interest for this analysis was SRH service utilization in the past 12 months (yes/no). Participants were asked which type of facility they visited in the past six months (public, private, pharmacy, or other non-governmental health facility), and which health services they received (general health, contraception, HIV test, STI test, pregnancy test, antenatal care, other reproductive health). Participants were also asked if they had ever had sexual intercourse (yes/no/no response), and when was the last time they had sexual intercourse (days, weeks, months, or years ago). We defined being sexually active as having had sexual intercourse in the past 12 months. For this analysis, we conceptualized barriers to and facilitators of SRH service utilization across the social-ecological model based on the literature (16,17), and selected corresponding variables from the questionnaire (Fig 1).

## Statistical analysis

Socio-demographic, structural, community, interpersonal, and individual variables were summarized overall and by SRH service utilization in the past 12 months. We conducted unadjusted logistic regression analyses to examine the relationship between each variable and SRH service utilization. Variables were selected for multiple logistic regression models based on bivariate analyses significance ($p < .05$). We fit four separate adjusted multiple logistical regression models for each of structural, health facility, community/interpersonal, and individual level factors. All models were adjusted for county, age, relationship status, orphan status, and religion. All analyses were conducted in RStudio Version 1.4.1106.

# Results

## Sociodemographic characteristics of participants

In total 1840 adolescent girls were included in the analysis. Over half (54%) of all participants were aged 15–17 years and 58% were from Homa Bay (Table 1). Most participants (60%) reported they were currently attending school, and few participants (21%) reported that they were engaged in work for money in the past six months. Overall, 60% (n = 1110) of

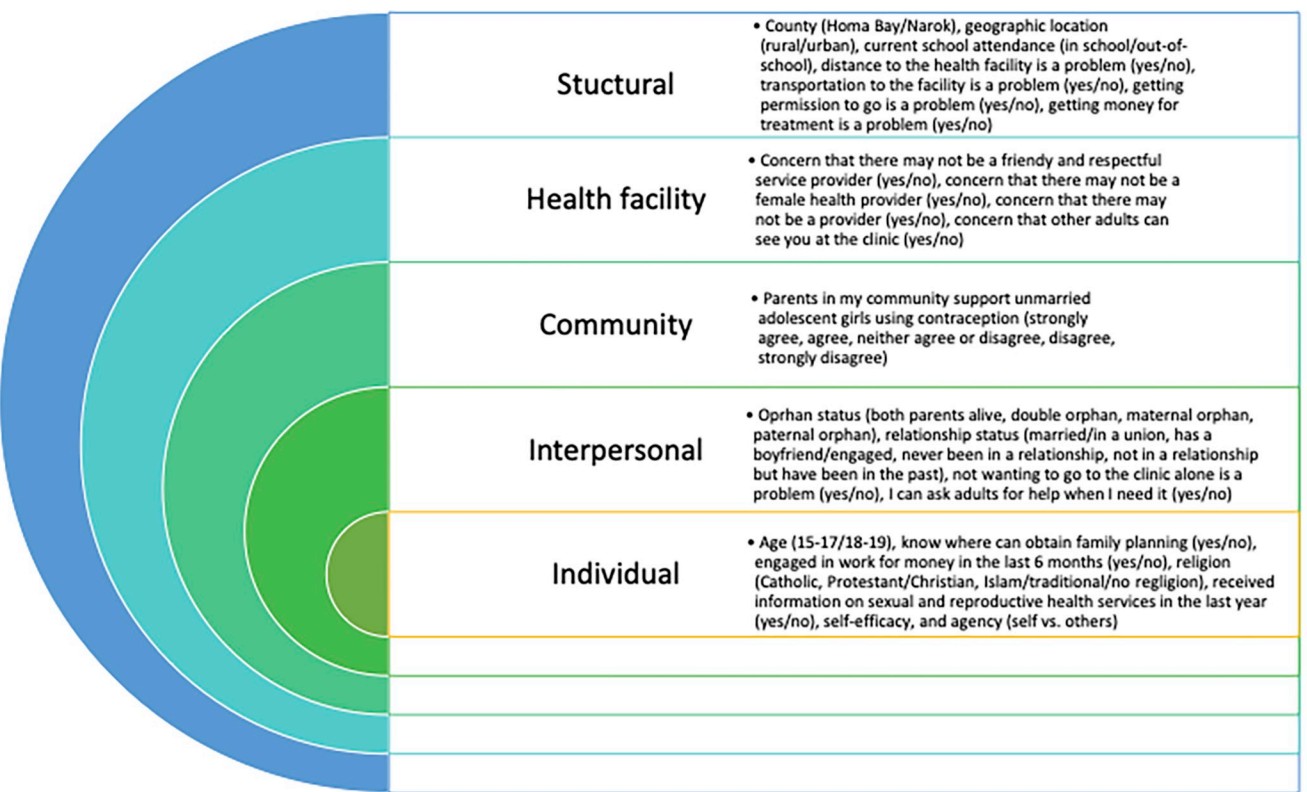

**Fig 1. Variables selected as factors influencing sexual and reproductive health utilization across the social-ecological model levels and included in analyse.**

participants reported having ever had sexual intercourse, of whom 83% (n = 920) reported being sexually active in the past 12 months.

## SRH service utilization

Over a third of all participants (36%, n = 664) reported visiting a health facility for SRH services in the past 12 months. Among participants who utilised SRH services, the majority (76%, n = 503) did so at a public facility, followed by a private facility (15%, n = 102), while the remaining (9%, n = 59) used other health facilities (e.g., pharmacy, non-governmental organization health facility, etc.). Most participants (49%) reported receiving an HIV test during their visits to a facility, followed by general care (29%), contraception (27%), and antenatal care (24%). Very few participants reported receiving an STI (5%) or pregnancy test (12%).

## Structural level factors influencing SRH utilization

SRH service utilization in the past 12 months varied by several structural factors (Table 2). A higher proportion of participants from Homa Bay (65%) reported SRH service utilization in the past 12 months in comparison to those in Narok (35%). A higher proportion (61%) of adolescent girls who indicated they were out-of-school reported using SRH services in the past 12 months in comparison to 39% of those in-school. About 21% (n = 393) of all participants reported that getting permission to go was a problem that prevented them from using SRH services. In the model examining structural factors influencing SRH utilization, after adjusting for age, relationship status, orphan status, and religion, adolescent girls in Homa Bay were 2.2

**Table 1. Socio-demographic characteristics of adolescent participants aged 15–19 in Kenya.**

| Socio-demographics | Total N = 1840 |
|---|---|
| **Age** | |
| 15–17 | 986 (53.6) |
| 18–19 | 854 (46.4) |
| **Current School Attendance** | |
| Yes | 1110 (60.3) |
| No | 700 (38.0) |
| **County** | |
| Narok | 779 (42.3) |
| Homa Bay | 1061 (57.7) |
| **Residence** | |
| Rural | 1055 (57.3) |
| Urban | 785 (42.7) |
| **Religion** | |
| Catholic | 389 (21.1) |
| Protestant/Christian | 1417 (77.0) |
| Islam/Tradition/No religion | 34 (1.8) |
| **Orphan Status** | |
| Both parents alive | 1222 (66.4) |
| Both parents deceased / don't know | 150 (8.2) |
| Maternal orphan | 82 (4.5) |
| Paternal orphan | 386 (21.0) |
| **Ethnic Group** | |
| Kikuyu | 131 (7.1) |
| Luo | 1077 (58.5) |
| Kisii | 78 (4.2) |
| Masai | 377 (20.5) |
| Kalenjin | 109 (5.9) |
| Other [a] | 68 (3.7) |
| **Relationship status** | |
| Currently married / in a union | 386 (21.0) |
| Has boyfriend/engaged | 679 (36.9) |
| Never been in a relationship | 535 (29.1) |
| Currently not in a relationship, but I have had a boyfriend in the past/divorced/separated/widowed | 239 (13.0) |
| **Engaged in work for money in the last 6 months** | |
| Yes | 388 (21.1) |
| No | 1452 (78.9) |
| **Ever had sexual intercourse** | |
| Yes | 1110 (60.3) |
| No | 730 (39.7) |

[a] Other (Embu = 1, Kamba = 11, Luhya = 35, Meru = 4, Mijikenda/Swahili = 1, Somali = 3, other = 13).

times (95% CI: 1.72–2.77) more likely to have used SRH services in the past 12 months in comparison to those in Narok. Similarly, out-of-school adolescent girls were 2.1 times (95% CI 1.60–2.82) more likely to use SRH services in the past 12 months than their school-going peers and those who did not require permission to go remained more likely to have used SRH services in the past 12 months (AOR: 1.37 95% CI: 1.04–1.82).

**Table 2. Structural factors associated with SRH utilization among adolescents aged 15–19 in Kenya.**

| Structural Factors | Total N = 1840 | SRH service utilization past 12 months (yes) n (%) n = 664 | SRH service utilization past 12 months (no) n (%) n = 1176 | Unadjusted Odds Ratio (95% CI) | Structural Model Adjusted[+] Odds Ratio (95%CI) |
|---|---|---|---|---|---|
| **County** | | | | | |
| Narok | 779 (42.3) | 232 (34.9) | 547 (46.5) | ref | ref |
| Homa Bay | 1061 (57.7) | 432 (65.1) | 629 (53.5) | **1.62 (1.33–1.97)** | **2.18 (1.72–2.77)** |
| **Geographical location** | | | | | |
| Rural | 1055 (57.3) | 300 (45.2) | 485 (41.2) | ref | - |
| Urban | 785 (42.7) | 364 (54.8) | 691 (58.8) | 0.85 (0.70–1.03) | - |
| **Current School Attendance** | | | | | |
| Out-of-school | 700 (38.0) | 397 (61.3) | 303 (26.1) | **4.48 (3.66–5.51)** | **2.12 (1.60–2.82)** |
| In school | 1110 (60.3) | 251 (38.7) | 859 (73.9) | Ref | ref |
| **Distance to health facility a problem** | | | | | |
| Yes | 341 (18.5) | 119 (17.9) | 222 (18.9) | ref | - |
| No | 1499 (81.5) | 545 (82.1) | 954 (81.1) | 1.07 (0.84–1.37) | - |
| **Transportation to the facility is a problem** | | | | | |
| Yes | 389 (21.1) | 146 (22.0) | 243 (20.7) | ref | - |
| No | 1451 (78.9) | 518 (78.0) | 933 (79.3) | 0.92 (0.73–1.17) | - |
| **Getting permission to go is a problem** | | | | | |
| Yes | 393 (21.4) | 106 (16.0) | 287 (24.4) | ref | ref |
| No | 1447 (78.6) | 558 (84.0) | 889 (75.6) | **1.70 (1.33–2.18)** | **1.37 (1.04–1.82)** |
| **Getting money for treatment is a problem** | | | | | |
| Yes | 658 (35.8) | 239 (36.0) | 419 (35.6) | ref | - |
| No | 1182 (64.2) | 425 (64.0) | 757 (64.4) | 0.98 (0.81–1.20) | - |

[+]Adjusted for: County, age, relationship status, orphan status, and religion.

### Health facility level factors influencing SRH utilization

Participants reported that several health facility level factors may prevent girls from getting SRH services (Table 3). These factors included concern that there may not be a friendly and respectful service provider (35%), that there may not be a female provider (29%), that there may not be any provider (29%), and that other adults may see them at the clinic (27%). In the adjusted model examining the association between health facility level factors and SRH utilization, none of these health facility level concerns were associated with SRH utilization.

### Community and interpersonal level factors influencing SRH utilization

Community and interpersonal level factors associated with SRH utilization are presented in Table 4. All community and interpersonal factors were associated with SRH service utilization in unadjusted analyses. In an adjusted model examining the association between community and interpersonal level factors and SRH utilization, participants who reported being able to ask adults for help when they needed it were more likely to report using SRH services in the

**Table 3. Health facility level factors associated with SRH utilization among adolescents aged 15–19 in Kenya.**

| Health Facility factors | Total N = 1840 | SRH service utilization past 12 months (yes) n (%) n = 664 | SRH service utilization past 12 months (no) n (%) n = 1176 | Unadjusted Odds Ratio (95% CI) | Health Facility Model Adjusted[+] Odds Ratio (95%CI) |
|---|---|---|---|---|---|
| **Concern that there may not be a friendly and respectful service provider** | | | | | |
| Yes | 636 (34.6) | 184 (27.7) | 452 (38.4) | ref | ref |
| No | 1204 (65.4) | 480 (72.3) | 724 (61.6) | **1.63 (1.33–2.01)** | 1.20 (0.91–1.60) |
| **Concern that there may not be female health provider** | | | | | |
| Yes | 536 (29.1) | 136 (20.5) | 400 (34.0) | ref | ref |
| No | 1304 (70.9) | 528 (79.5) | 776 (66.0) | **2.00 (1.60–2.51)** | 1.24 (0.91–1.68) |
| **Concern that there may not be a provider** | | | | | |
| Yes | 539 (29.3) | 169 (25.5) | 370 (31.5) | ref | ref |
| No | 1301 (70.7) | 495 (74.5) | 806 (68.5) | **1.34 (1.09–1.67)** | 1.02 (0.77–1.35) |
| **Concern that other adults can see you at the clinic** | | | | | |
| Yes | 500 (27.2) | 137 (20.6) | 363 (30.9) | ref | ref |
| No | 1340 (72.8) | 527 (79.4) | 813 (69.1) | **1.72 (1.37–2.16)** | 1.11 (0.82–1.49) |

[+]Adjusted for: County, age, relationship status, orphan status, and religion.

past 12 months (AOR: 1.98, 95% CI: 1.09–3.78) compared to those who could not ask adults for help. While adolescents who had a boyfriend/were engaged, those who were in a relationship in the past, and those who had never been in a relationship were 68%, 73%, and 92%, less likely to have used SRH services in the past 12 months, respectively compared to those who were married or in a union.

## Individual factors influencing SRH utilization

Table 5 presents unadjusted and adjusted estimates of the association between individuals level factors and SRH service utilization. Several individual level factors were associated with SRH service utilization in unadjusted analyses. In an adjusted model examining individual level factors, those who were aged 18–19 years were 1.72 (95% CI: 1.35–2.20) times more likely to have used SRH services in the past 12 months in comparison to participants aged 15 to 17 years. Adolescents who reported engaging in work for money in the past six months remained 1.41 (95% CI: 1.07–1.84) times more likely to have used SRH services in comparison to those who did not work. Having knowledge about where to obtain family planning remained significantly associated with SRH service utilization (AOR = 2.48 95% CI: 1.74–3.57). Similarly, participants who reported receiving information on SRH services in the past year had a 1.44 (95% CI:1.15–1.80) times odds of using SRH services in comparison to those who did not.

## Discussion

The present analysis identified low levels of SRH service utilization among participants in this evaluation, yet many of the adolescent girls surveyed reported ever having sex and more than

**Table 4. Community and interpersonal factors associated with SRH utilization among adolescents aged 15–19 in Kenya.**

| Community and Interpersonal Factors | Total N = 1840 | SRH service utilization past 12 months (yes) n (%) n = 664 | SRH service utilization past 12 months (no) n (%) n = 1176 | Unadjusted Odds Ratio (95% CI) | Community & Interpersonal Model Adjusted[+] Odds Ratio (95%CI) |
|---|---|---|---|---|---|
| **Community Factors** | | | | | |
| **Parents in my community support unmarried girls using contraception** | | | | | |
| Strongly Agree / Agree | 390 (21.2) | 160 (24.1) | 230 (19.6) | **1.34 (1.06–1.69)** | 1.18 (0.91–1.53) |
| Neither Agree or Disagree | 208 (11.3) | 79 (11.9) | 129 (11.0) | 1.18 (0.87–1.59) | 1.03 (0.73–1.44) |
| Disagree / Strongly Disagree | 1242 (67.5) | 425 (64.0) | 817 (69.5) | ref | ref |
| **Interpersonal Factors** | | | | | |
| **Orphan Status** | | | | | |
| Both parents alive | 1222 (66.4) | 391 (58.9) | 831 (70.7) | ref | ref |
| Double orphan / vital status unknown | 150 (8.2) | 73 (11.0) | 77 (6.5) | **2.01 (1.43–2.83)** | 1.20 (0.81–1.77) |
| Maternal orphan | 82 (4.5) | 32 (4.8) | 50 (4.3) | 1.36 (0.85–2.14) | 1.17 (0.70–1.93) |
| Paternal orphan | 386 (21.0) | 168 (25.3) | 218 (18.5) | **1.64 (1.29–2.07)** | 1.27 (0.97–1.65) |
| **Relationship Status** | | | | | |
| Currently married / in a union | 386 (21.0) | 254 (38.3) | 132 (11.2) | **ref** | **ref** |
| Has boyfriend/engaged | 679 (36.9) | 256 (38.6) | 423 (36.0) | **0.32 (0.24–0.41)** | **0.35 (0.26–0.46)** |
| Never been in a relationship | 535 (29.1) | 72 (10.8) | 463 (39.4) | **0.08 (0.06–0.11)** | **0.12 (0.08–0.18)** |
| Not in a relationship, but had bf in the past /divorced/separated/widowed | 239 (13.0) | 82 (12.3) | 157 (13.4) | **0.27 (0.19–0.38)** | **0.33 (0.23–0.47)** |
| **Not wanting to go to the clinic alone is a problem** | | | | | |
| Yes | 489 (26.6) | 134 (20.2) | 355 (30.2) | ref | ref |
| No | 1351 (73.4) | 530 (79.8) | 821 (69.8) | **1.71 (1.37–2.15)** | 1.13 (0.88–1.46) |
| **I can ask adults for help when I need it** | | | | | |
| Yes | 1758 (95.5) | 649 (97.7) | 1109 (94.3) | **2.61 (1.52–4.79)** | **1.98 (1.09–3.78)** |
| No | 82 (4.5) | 15 (2.3) | 67 (5.7) | **ref** | **ref** |

[+]Adjusted for: County, age, relationship status, orphan status, and religion.

half of them indicated that they were sexually active in the past 12 months. This suggests that there remains a substantial need to strengthen the delivery of AYFS and improve access to SRH services for this population. In alignment with findings from other sub-Saharan African contexts [16], our analysis found several factors across the social-ecological model that were associated with SRH service utilization that should be targeted with evidence-based policies, practices, and interventions to increase access to and utilization of these services in Kenya.

At the structural level, our analysis identified that adolescent girls in Homa Bay, those who were out-of-school, and those who responded that getting permission to go for SRH services was not a problem, were significantly more likely to have used SRH services in the past 12 months. Political, economic, social-cultural, and other contextual factors at the county level are likely to influence adolescent SRH utilization. First, the delivery of health is devolved to county level governments in Kenya, which may differentially influence health facility financing, distribution of the health workforce, and resources across counties, thereby leading to inequities across the healthcare system [28,29]. In turn, this may influence the delivery of AYFS, resulting in lower SRH utilization in Narok versus Homa Bay. Second, social-cultural

**Table 5. Individual level factors associated with SRH utilization among adolescents aged 15–19 in Kenya.**

| Individual level factors | Total N = 1840 | SRH service utilization past 12 months (yes) n (%) N = 664 | SRH service utilization past 12 months (no) n (%) N = 1176 | Unadjusted Odds Ratio (95% CI) | Individual Model Adjusted[+] Odds Ratio (95%CI) |
|---|---|---|---|---|---|
| **Age** | | | | | |
| 15–17 | 986 (53.6) | 225 (33.9) | 761 (64.7) | Ref | ref |
| 18–19 | 854 (46.4) | 439 (66.1) | 415 (35.3) | **3.58 (2.93–4.38)** | **1.72 (1.35–2.20)** |
| **Do you know of a place where you can obtain a method of family planning?** | | | | | |
| Yes | 1470 (79.9) | 619 (93.2) | 851 (72.4) | **5.25 (3.82–7.38)** | **2.48 (1.74–3.57)** |
| No | 370 (20.1) | 45 (6.8) | 325 (27.6) | ref | ref |
| **Engaged in work last 6 months for money** | | | | | |
| Yes | 388 (21.1) | 191 (28.8) | 197 (16.8) | **2.00 (1.60–2.52)** | **1.41 (1.07–1.84)** |
| No | 1452 (78.9) | 473 (71.2) | 979 (83.2) | ref | ref |
| **Religion** | | | | | |
| Catholic | 389 (21.1) | 120 (18.1) | 269 (22.9) | ref | ref |
| Protestant/Christian | 1417 (77.0) | 535 (80.6) | 882 (75.0) | **1.36 (1.07–1.73)** | 1.29 (0.98–1.70) |
| Muslim/Tradition/No religion | 34 (1.8) | 9 (1.3) | 25 (22.1) | 0.81 (0.35–1.72) | 0.77(0.31–1.78) |
| **Received information on sexual and reproductive health services in the last one year** | | | | | |
| Yes | 966 (52.5) | 407 (61.3) | 559 (47.5) | **1.75 (1.44–2.12)** | **1.44 (1.15–1.80)** |
| No | 874 (47.5) | 257 (38.7) | 617 (52.5) | ref | ref |
| **Self-Efficacy** | | | | | |
| Mean (SD) | 7.43 (1.84) | 7.45 (1.79) | 7.43 (1.87) | 1.01 (0.96–1.06) | - |
| **Agency** | | | | | |
| **A. How to spend money** | | | | | |
| Self | 357 (19.4) | 188 (28.3) | 169 (14.4) | **2.35 (1.86–2.98)** | 1.16(0.87–1.55) |
| Others | 1483 (80.6) | 476 (71.7) | 1007 (85.6) | ref | ref |
| **B. Whether or not you work for pay** | | | | | |
| Self | 730 (39.7) | 330 (49.7) | 400 (34.0) | **1.92 (1.58–2.33)** | 1.11(0.85–1.45) |
| Others | 1110 (60.3) | 334 (50.3) | 776 (66.0) | ref | ref |
| **C. Whether or not you go to school** | | | | | |
| Self | 762 (41.4) | 326 (49.1) | 436 (37.1) | **1.64 (1.35–1.99)** | 1.04 (0.81–1.33) |
| Others | 1078 (58.6) | 338 (50.9) | 740 (62.9) | ref | ref |
| **D. Who decides/decided when you would get married** | | | | | |
| Self | 1267 (68.9) | 512 (77.1) | 755 (64.2) | **1.89 (1.51–2.34)** | 1.11(0.85–1.46) |
| Others | 573 (31.1) | 152 (22.9) | 421 (35.8) | ref | ref |
| **E. Who I can have as friends** | | | | | |
| Self | 1702 (92.5) | 625 (94.1) | 1077 (91.6) | **1.47 (1.01–2.18)** | 0.99 (0.64–1.55) |
| Others | 138 (7.5) | 39 (5.9) | 99 (8.4) | ref | ref |

[+]Adjusted for: County, age, relationship status, orphan status, and religion.

and gender norms affecting SRH utilization across these two counties may be disparate, as the ethnic composition of the populations and participants sampled from these two counties were different. Finally, Homa Bay county has a high prevalence of HIV [10], and several SRH-related interventions targeting adolescent girls, such as the USAID DREAMS initiative to reduce HIV in adolescent girls [30,31], have been implemented in this setting, which may have had an impact on adolescent girls' higher SRH utilization in this county. Differential SRH utilization across counties by adolescent girls in this analysis points to the need to augment and strengthen Kenya's universal health coverage rollout, with a special focus on adolescent SRH [32]. Furthermore, inequities in the delivery and utilization of health services across counties [28], points to the need to collect sub-national and granular geographic data on adolescent SRH outcomes and service utilization by facility to monitor progress towards achieving SDGs and informing county-level health planning.

Being out-of-school was independently associated with adolescent girls' SRH utilization, after adjusting for important factors such as age, county, and relationship status. School attendance is an important structural determinant of adverse SRH outcomes. Adolescent girls who are out-of-school in this context, are more likely to have engaged in sex, have had unprotected sex, commenced childbearing, and experienced unintended pregnancies [5], which likely contributed to their SRH utilization. At the same time, in-school adolescents in Kenya, generally lack comprehensive sexuality education, are primarily taught abstinence, and therefore may have minimal knowledge about SRH, and have an absence of awareness about the availability of and how to access adolescent-friendly SRH services [33,34], thus reducing SRH utilization. These findings point to the essential and urgent need to strengthen comprehensive sexuality education in the school system in Kenya [34].

Finally at the structural level, the need for parental or partner consent is a well-established barrier to SRH utilization [2,18]. The national guidelines for AYFS and the national family planning guidelines outline that parental consent is not a requirement of adolescents receiving SRH services [2,35]. Nevertheless, just under a quarter of adolescent girls surveyed in this study reported that getting permission to go for SRH services was a problem. While parental/guardian consent may not be needed at health facilities, parents or partners may still prevent adolescent girls from seeking SRH services. This may reflect the highly gender inequitable and social-cultural context in Kenya, where patriarchal values are the norm, resulting in adolescent girls requiring parental or partner permission [36]. Structural interventions to improve SRH utilization among adolescents need to extend beyond strengthening the health system, and incorporate initiatives to address broad social-cultural and gender norms.

At the community and interpersonal level, most adolescents reported that parents in their communities did not support unmarried adolescent girls using contraception, which is in alignment with the social-cultural norms in this context. In Kenya, social-cultural norms construct adolescent sexuality and contraceptive use as taboo and inappropriate for unmarried adolescents [21–24]. Our analysis further supports this, as unmarried adolescents were significantly less likely to have used SRH services in the past 12 months. While adolescents in Kenya discuss stigma, and fear of parental judgment, family dishonor, and shame related to SRH issues [22,24], parental support and the ability to ask adults for help in the present analysis was associated with SRH utilization among participants. Interventions to destigmatize adolescent sexuality and contraceptive use in the community and to improve communication about sensitive topics between parents and adolescents are vital to improving sexual health and SRH service utilization among adolescent girls in this context.

Lastly our analysis identified key individual level factors influencing SRH utilization that are important areas for intervention. Participants who reported receiving information on SRH services were significantly more likely to use SRH services in our adjusted analysis. In a related

manner, knowledge about where to obtain a method of family planning was highly associated with SRH utilization. This suggests that SRH health promotion and comprehensive sexuality education are fundamental components of improving adolescent girls' utilization of SRH services in combination with other interventions, policies, and practices addressing structural, community, and interpersonal barriers to care.

While this study identified several factors across the social-ecological model that were associated with SRH utilization among adolescent girls in Kenya, it is not without limitations. Our study did not collect data on household level socio-economic status, which may be an important factor associated with adolescent girls' SRH utilization Additionally, extensive data on health facility and provider characteristics or community level factors that may influence adolescent SRH utilization was not ascertained. Health facility level factors, such as healthcare provider attitudes and stigma and discrimination, feature prominently as barriers to SRH utilization in several studies [16,17,20,37]. Likewise community level social-cultural norms, stigma, and discrimination are known barriers to adolescent SRH utilization [21–24,37]. Therefore, our results are limited in understanding the impact of health facility and community levels factors on adolescent SRH utilization. Next, a proportion (29%) of adolescents who responded that they used SRH services, indicated that they received 'general care'. It may be that given that sexual health is a sensitive topic, adolescent girls did not want to disclose the type of SRH service they sought, or they misunderstood the nature of SRH care, which would mean our analysis overestimated the number or adolescent girls who used SRH services in the last 12 months. Finally, the study did not include younger adolescents aged 10–14 years, and therefore the results may not be representative of factors influencing SRH utilization among younger adolescent girls. Notwithstanding these limitations, this analysis used data from a robust sample of adolescent girls across two counties in Kenya. Further we were able to conduct analyses to look across social-ecological levels to identify key factors associated with SRH utilization among adolescent girls that can be targeted in future interventions.

## Conclusion

This analysis found low levels of SRH utilization among adolescent girls in Homa Bay and Narok counties in Kenya. Given that adolescent girls continue to experience a substantial burden of adverse SRH outcomes, there is an urgent need to improve access to and utilization of AYFS in this context. Overall, our findings demonstrate the need for interventions, policies, and practices to be implemented across the structural, health facility, community, interpersonal, and individual levels to comprehensively support adolescent girls to use SRH services.

## Acknowledgments

We would like to acknowledge the study participants: adolescents, parents, and the community health volunteers, who provided essential information for this study. We are also grateful to the ITH research team including the research assistants who contributed to the implementation of the ITH baseline survey.

## Author Contributions

**Conceptualization:** Lonnie Embleton, Paula Braitstein, Erica Di Ruggiero, Clement Oduor, Yohannes Dibaba Wado.

**Formal analysis:** Lonnie Embleton.

**Funding acquisition:** Yohannes Dibaba Wado.

**Investigation:** Clement Oduor, Yohannes Dibaba Wado.

**Methodology:** Yohannes Dibaba Wado.

**Supervision:** Yohannes Dibaba Wado.

**Writing – original draft:** Lonnie Embleton.

**Writing – review & editing:** Paula Braitstein, Erica Di Ruggiero, Clement Oduor, Yohannes Dibaba Wado.

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
