## [Decision Letter · Decision Letter 0]

31 Oct 2022

PGPH-D-22-01261

Sexual and reproductive health service utilization among adolescent girls in Kenya: a cross-sectional analysis

Dear Dr. Di Ruggiero,

Thank you for submitting your manuscript to PLOS Global Public Health. After careful consideration, we feel that it has merit but does not fully meet PLOS Global Public Health’s publication criteria as it currently stands. Therefore, we invite you to submit a revised version of the manuscript that addresses the points raised during the review process.

We look forward to receiving your revised manuscript.

Kind regards,

Anil Gumber, Ph.D.

Academic Editor

Journal Requirements:

a. Please clarify all sources of funding (financial or material support) for your study. List the grants (with grant number) or organizations (with url) that supported your study, including funding received from your institution. 

b. State the initials, alongside each funding source, of each author to receive each grant.

c. State what role the funders took in the study. If the funders had no role in your study, please state: “The funders had no role in study design, data collection and analysis, decision to publish, or preparation of the manuscript.”

d. If any authors received a salary from any of your funders, please state which authors and which funders.

2. Please provide separate figure files in .tif or .eps format only and remove any figures embedded in your manuscript file. Please also ensure that all files are under our size limit of 10MB.

3. In the online submission form, you indicated that "Data is available upon reasonable request". All PLOS journals now require all data underlying the findings described in their manuscript to be freely available to other researchers, either 1. In a public repository, 2. Within the manuscript itself, or 3. Uploaded as supplementary information.

Additional Editor Comments (if provided):

An important subject has been covered by the authors. However there are some methodological issues and analysis need further strengthening. The study has tried to identify access and utilisation rates across two regions; instead they should have focus on the determinants and association in rural and urban areas separately. We understand the access to healthcare facilities are much better in urban than in rural areas; however, a surprise findings/differences do appear in rural and urban areas but the utilisation of sexual and reproductive services was much greater in rural than urban adolescent girls. Certain statistics are provided for 10-19 age groups whereas the analysis focused on 15-19 age group; one needs to re-do those statistics relevant for 15-19 age group. Several tables no statistical significance tests were attempted. Authors need to present all tables/analysis separately for rural and urban areas. The discussion needs to be improved in the context of rural-urban differences in sexual health seeking behaviour.

Reviewers' comments:

Reviewer's Responses to Questions

**Comments to the Author**

1. Does this manuscript meet PLOS Global Public Health’s publication criteria? Is the manuscript technically sound, and do the data support the conclusions? The manuscript must describe methodologically and ethically rigorous research with conclusions that are appropriately drawn based on the data presented.

Reviewer #1: Yes

Reviewer #2: Yes

2. Has the statistical analysis been performed appropriately and rigorously?

Reviewer #1: Yes

Reviewer #2: Yes

3. Have the authors made all data underlying the findings in their manuscript fully available (please refer to the Data Availability Statement at the start of the manuscript PDF file)?

Reviewer #1: Yes

Reviewer #2: No

4. Is the manuscript presented in an intelligible fashion and written in standard English?

Reviewer #1: Yes

Reviewer #2: Yes

5. Review Comments to the Author

Reviewer #1: very well written manuscript with sound statistical analysis. I have made comments and tracked them where the authors can see and address the comments. I am happy to take a look at the manuscript once the authors have addressed the comments

Reviewer #2: This paper presents the results of a cross sectional survey investigating associations between adolescents’ sexual and reproductive health service utilization and structural, health facility, community, interpersonal, and individual level factors in Kenya. In general it is robust, thorough, has a strong methodology and interesting results which are well interpreted.

The below comments are suggested to tighten up the paper, with a few considerations to add to the discussion / limitations.

P6 Study setting highest adolescent pregnancy .. please add age cut off – is it 15-18y,10-18y, 15-19y, 10-19y

P7 Study participants, sampling – please clarify for sub-county, wards, and villages, how you sampled. Similarly how was random sampling done of the adolescent girls per household

P7 What was the time lag between the generation of the sampling frame for adolescent girls aged 15-19y and the adolescent survey? (i.e. some of girls reached 20y?

P7 Consider noting the percent refusals e.g., 57 (3%) were refusals or unavailable.

P7-8 No mention is made of consenting of full orphans who were under age 18 – did they all live with guardians and have guardian consent or were some emancipated minors?

P8 In line 185 note typo ‘each selected household’ (no s)

P8 Kiswahili as local language is a surprise – in Homa Bay Luo the local language and Table 1 indicates 58% of the survey population were Luo – were there any issues using Kiswahili? Also, were there are any illiterate out-of-school girls who required face to face interview instead of completing electronic tablet for the survey? In rereading, I could not see if the survey was self-completed or by interviewer (maybe I missed).

P9 – any attempt to differentiate sexual activity in past 12 months between ‘willing’ and ‘unwilling’ (forced, tricked, etc)

P9 – shame there was nothing on household economics (as a household survey, and nested in larger study, was there a parallel socio-economic status indicator at household level – one would imagine relative poverty would be an important measure to include in analyses.

Results

P9-10/Table 1 – orphan status seems to include unknown as well as both parents died; who are the ‘unknowns’ and what proportion are they of the 150 ‘double orphans’ category - wondering if these have a diluting effect on any analyses?

P11 Authors noted that about half of participants reported their SRH service was receiving an HIV test during their visits to a facility, followed by general care (29%), contraception (27%), and antenatal care (24%). Have the authors explored whether there were significant differences within their model e.g. at interpersonal, personal level for these key services – would it be worth having as a supplementary table if there were differences ?

Adjusted models by relationship status to control for exposure to SRH harms, did authors also look at reported sex / no sex, with the latter ~40% of the population (n=730)?

Discussion

Well written and thorough. Authors may wish to add that they were unable to include socio-economic status of their household to the limitations paragraph.

6. PLOS authors have the option to publish the peer review history of their article (what does this mean?). If published, this will include your full peer review and any attached files.

**Do you want your identity to be public for this peer review?** For information about this choice, including consent withdrawal, please see our Privacy Policy.

Reviewer #1: **Yes: **Erick Kiprotich Yegon

Reviewer #2: No

---

## [Editor Report · Decision Letter 1]

29 Dec 2022

Sexual and reproductive health service utilization among adolescent girls in Kenya: a cross-sectional analysis

PGPH-D-22-01261R1

Dear Dr. Di Ruggiero,

We are pleased to inform you that your manuscript 'Sexual and reproductive health service utilization among adolescent girls in Kenya: a cross-sectional analysis' has been provisionally accepted for publication in PLOS Global Public Health.

Best regards,

Anil Gumber, Ph.D.

Academic Editor